# Emerging Roles for G Protein-Coupled Estrogen Receptor 1 in Cardio-Renal Health: Implications for Aging

**DOI:** 10.3390/biom12030412

**Published:** 2022-03-07

**Authors:** Ravneet Singh, Victoria L. Nasci, Ginger Guthrie, Lale A. Ertuglu, Maryam K. Butt, Annet Kirabo, Eman Y. Gohar

**Affiliations:** 1Division of Nephrology and Hypertension, Vanderbilt University Medical Center, Medical Research Building IV, Nashville, TN 37232, USA; ravneet.singh@vumc.org (R.S.); victoria.nasci@vumc.org (V.L.N.); 2Division of Nephrology, Department of Medicine, University of Alabama at Birmingham, Birmingham, AL 35233, USA; glg1280@uab.edu (G.G.); maryam27@uab.edu (M.K.B.); 3Division of Clinical Pharmacology, Vanderbilt University Medical Center, Nashville, TN 37232, USA; lale.ertuglu@vumc.org (L.A.E.); annet.kirabo@vumc.org (A.K.)

**Keywords:** GPR30, kidney, heart, postmenopausal women, RAAS, endothelin-1, nitric oxide, ROS

## Abstract

Cardiovascular (CV) and renal diseases are increasingly prevalent in the United States and globally. CV-related mortality is the leading cause of death in the United States, while renal-related mortality is the 8th. Despite advanced therapeutics, both diseases persist, warranting continued exploration of disease mechanisms to develop novel therapeutics and advance clinical outcomes for cardio-renal health. CV and renal diseases increase with age, and there are sex differences evident in both the prevalence and progression of CV and renal disease. These age and sex differences seen in cardio-renal health implicate sex hormones as potentially important regulators to be studied. One such regulator is G protein-coupled estrogen receptor 1 (GPER1). GPER1 has been implicated in estrogen signaling and is expressed in a variety of tissues including the heart, vasculature, and kidney. GPER1 has been shown to be protective against CV and renal diseases in different experimental animal models. GPER1 actions involve multiple signaling pathways: interaction with aldosterone and endothelin-1 signaling, stimulation of the release of nitric oxide, and reduction in oxidative stress, inflammation, and immune infiltration. This review will discuss the current literature regarding GPER1 and cardio-renal health, particularly in the context of aging. Improving our understanding of GPER1-evoked mechanisms may reveal novel therapeutics aimed at improving cardio-renal health and clinical outcomes in the elderly.

## 1. Introduction

### 1.1. Cardio-Renal Health

Cardiovascular (CV)-related events are the leading causes of mortality in the United States (US) and worldwide [1,2]. Renal disease is often both a cause and result of CV disease (CVD) [3] and is among the top 10 leading causes of death in the US and globally [1,2]. In 2020, heart disease was the leading cause of death in the US, accountable for 696,962 deaths, and kidney disease was the tenth cause, responsible for 52,547 deaths out of 3,383,729 total deaths [2]. Despite the advanced therapies available, CV and renal disease remain increasingly prevalent, and the incidence of mortality from both continue to rise in the US and globally [4,5]. Thus, there is a need to advance our understanding of disease pathways that could provide new therapeutics to improve cardio-renal clinical outcomes.

### 1.2. Aging as a Contributor to Cardio-Renal Disease

The overall life expectancy globally has increased by more than 6 years, from 66.8 in 2000 to 73.4 years in 2019 [6]. Moreover, the elderly population in the US (>65 years of age) is projected to nearly double from 52 million in 2018 to 95 million by 2060, or approximately 23% of the overall population [7]. The elderly population, when compared to the younger, has a higher incidence of CVD (26% in aged ≥70 years vs. 1% in those aged 20–39 years) and chronic kidney disease (CKD) (38.1% in aged ≥65 years vs. 6% in those aged 18–44 years) [8,9]. Taken together, these factors suggest that the societal burden of CVD and CKD will continue to grow.

The increase in disease prevalence is in part due to pathophysiological changes in structure and function due to aging [10,11]. Within the CV system, aging is associated with an increase in autophagy, artery stiffness, endothelial dysfunction, plaque formation, vascular remodeling, and increased cardiac fibrosis [12,13,14,15,16]. Furthermore, cardio-protective pathways and sinoatrial function in cardiomyocytes slow down as aging progresses [17]. Within the renal system, aging is associated with reduced glomerular function, kidney cortical volume, glomerular filtration rate and number of nephrons with increased nephrosclerosis [18,19,20,21,22,23,24]. These changes associated with aging are of clinical significance with negative impacts on cardio-renal health. Therefore, it is important to expand research focusing on cardio-renal health, particularly in an aging population, to advance health outcomes.

### 1.3. Sex and Sex Hormones as Contributors to Cardio-Renal Disease

In addition to the contribution of aging, other factors can impact cardio-renal health in elderly. For instance, CVD is more prevalent in young men than in age-matched women [25,26,27]. CVD increases with age in both men and women, but increases are more pronounced in women, particularly with menopause [25,26]. However, women tend to be underdiagnosed and they ultimately suffer more CV-related complications compared to men [4,26,27,28]. While the impact of sex and sex hormones on prevalence and progression of CVD and CKD is still being investigated, it remains evident that sex should be considered during exploration of cardio-renal mechanisms.

Furthermore, evidence suggests a role for gonadal hormones and their receptors in mediating these gender-dependent differences in CVD and CKD prevalence and progression. Importantly, the level of circulating sex steroid hormones declines during aging [29,30]. The age-related decline in circulating testosterone in men has been linked to adverse CV outcomes including atherosclerosis, coronary artery disease and other events that ultimately lead to an increased CV-related mortality [31,32,33,34]. There is conflicting evidence regarding whether testosterone supplementation improves CV outcomes. Several studies suggest that while testosterone supplementation can improve metabolic changes, it may also increase blood pressure, thus increasing cardio-renal risk [35,36,37].

Multiple lines of study have reported that estrogen deficiency increases CV risk. Aged females undergo sex hormone changes associated with menopause, primarily the loss of endogenous estrogen production from the ovaries [38,39]. Previous studies reported that estrogen deficiency leads to a rapid and sustained increase in blood pressure [38,39,40,41] and outlined estrogen’s protective role against high blood pressure [38]. Thus, growing evidence implicates a central role for sex steroid hormones within the CV and renal systems especially in aged individuals. A better understanding of mechanisms involved in sex differences in disease and the impact of age are therefore imperative to improve cardio-renal clinical outcomes through the development of novel therapeutics.

## 2. GPER1 as a Regulator for Cardio-Renal Health

One potential novel mediator of sex differences in cardio-renal health is G protein-coupled estrogen receptor 1 (GPER1). GPER1, previously known as G protein-coupled receptor 30 (GPR30), is a seven transmembrane G protein-coupled receptor involved in estrogen-mediated signaling [42,43,44]. GPER1 is expressed widely throughout various organ systems in the body, including the nervous system [45,46,47,48,49], reproductive system [43,46,47,49,50,51], gastrointestinal system [47,49], musculoskeletal system [47,48,49,52], digestive system [48,53,54], and importantly in the cardiovascular and renal systems [43,47,49,55,56].

The purpose of this review is to outline current understanding of GPER1-mediated actions and signaling pathways that may play a role in cardio-renal disease. In addition, we provide evidence of the potential novel role of GPER1 in the preservation of cardiovascular and renal health, particularly during aging.

### 2.1. Expression of GPER1 within the Cardiovascular and Renal Systems

Within the cardiovascular system, GPER1 is expressed in vascular beds, in both endothelial and vascular smooth muscle cells (VSMC) [48,49,55,57,58,59,60]. In the heart, GPER1 is expressed in mast cells [61], cardiac fibroblasts, coronary artery endothelial cells [62], coronary artery VSMCs [63], and cardiomyocytes [64,65]. Within the kidney, GPER1 is expressed in a variety of tubular epithelial cells in both the cortex and medulla [66,67]. Additionally, GPER1 is expressed in renal interlobular arteries [68] and mesangial cells [69].

GPER1 mRNA expression in kidney is higher in female rats than male rats [56,70]. Importantly, aging increases myocardial GPER1 mRNA expression in male and female mice [71]. However, renal and aortic GPER1 mRNA expression is increased with aging in female, but not male, mice [71], suggesting a potential female-specific role for GPER1 in regulating renal and vascular function during aging (Figure 1).

### 2.2. Pharmacological and Genetic Tools to Study GPER1 Function

GPER1-related research has rapidly advanced following the discovery of a selective agonist and antagonists. G-1, a non-steroidal compound developed in 2007 by Bologa and colleagues, is a selective GPER1 agonist (Figure 2) [72]. Studies utilizing GPER1 knockout (KO) mice have shown that G-1 activates GPER1 mediated signaling in wild type, but not KO mice [60,73,74]. Furthermore, in vitro studies have shown no binding affinity for G-1 to estrogen receptors α or β (ERα and ERβ) [72,75]. G-1 modulates signaling pathways involved in the regulation of intracellular calcium [Ca^2+^], phosphoinositide 3-kinase (PI3Ks), extracellular signal-related kinases (ERKs) and cyclic adenosine monophosphate (cAMP) [76,77].

GPER1 antagonists include G-15 and G-36 (Figure 2). G-15 has been classified as a selective GPER1 antagonist with little binding to estrogen receptors (ERα and ERβ) and high binding affinity to GPER1 [78]. In COS7 and SKBr3 cells, G-15 inhibited G-1 and estradiol (E_2_)-mediated Ca^2+^ mobilization and blocked activation of PI3K signaling through GPER1, but, despite slight binding, not ERα and ERβ [78]. G-36 has also been classified as a selective GPER1 antagonist. G-36 inhibited G-1 and E_2_ stimulated Ca^2+^ mobilization and PI3K signaling through GPER1 and not ERα and ERβ in COS7 and SKBr3 cells similarly to G-15 [79]. G-36 was shown to bind more specifically to GPER1 than G-15. G-15 has been shown to display slight agonism towards ERα and ERβ activating estrogen response elements ~15%. G-36 on the contrary, displayed only ~5% activation of estrogen response elements displaying a more specific binding and specific antagonism of GPER1 signaling [79].

Overall, G-1, G-15, and G-36 allow for important exploration of GPER1 involvement in other signaling mechanisms [79]. Further, the utilization of GPER1 KO mice has provided another powerful tool to identify GPER1 actions under physiological and pathophysiological conditions.

### 2.3. Mediators of GPER1 Actions within the Cardiovascular and Renal Systems

The exact signaling mechanisms and transduction pathways of GPER1 are not completely understood, however, they are likely to be influenced by cell type, sex, site of action, and relative levels to other estrogen receptors. In this section, we will elaborate on the interaction between GPER1 signaling and other signaling pathways that play critical roles in the maintenance of cardio-renal health including the renin-angiotensin aldosterone system (RAAS), endothelin-1 (ET-1), nitric oxide (NO), reactive oxygen species (ROS), mitochondria, immunity, and inflammation.

#### 2.3.1. Renin-Angiotensin Aldosterone System

The RAAS plays an important role in the cardio-renal system through regulation of blood pressure, fluid, and electrolyte homeostasis. The RAAS is activated in response to a sensed decrease in arterial blood pressure, decreased salt delivery to the distal convoluted tubule, and/or beta-adrenergic activation via sympathetic nerve input [80,81]. Renin, which is stored in the juxtaglomerular cells of the kidney, is released following stimulus where it then converts angiotensinogen, from the liver, into angiotensin I. Angiotensin I is then converted into angiotensin II (ANG II) by angiotensin converting enzyme (ACE). ANG II stimulates an increase in blood pressure through several mechanisms, including vasoconstriction and sodium and water retention through aldosterone and antidiuretic hormone (ADH) [80,81]. Aldosterone effects sodium handling by increasing the expression of epithelial sodium channel (ENaC) and sodium-potassium ATPase (Na/K ATPase) on the cell membrane in the collecting duct of the nephron [80,81]. ADH affects water handling by inserting aquaporins on the cell membrane in the collecting duct of the nephron [80,81].

Dysfunction of the RAAS has been implicated in the pathophysiology of cardiovascular and kidney diseases. Importantly, there are sex differences in RAAS expression. Plasma renin levels are lower in females than males [82,83]. In the hypertensive mRen2.Lewis rat, plasma renin levels are higher in males than females [84]. Plasma ACE levels are lower in young women than men, [85] however, there are no differences in these levels between genders in the middle and older ages [83,86]. Interestingly, oral hormone replacement therapy in postmenopausal women reduces ACE levels [83,87]. ANG II levels in normotensive premenopausal women are similar to those in men [83]. Data in the hypertensive mRen2.Lewis rat suggest that in pathological settings ANG II is higher in males compared to females [84]. Plasma aldosterone levels are reduced in premenopausal women compared to men and this difference is lost with menopause [83,86].

More importantly, there is a sex difference in the response to ANG II. Several studies show female mice have a reduced response to ANG II-induced blood pressure increase [88,89]. Interestingly, a study by Sartori-Valinotti et al. showed that when treated with an ACE inhibitor, female Sprague Dawley rats had a more robust response to ANG II-induced blood pressure increase than males [90]. A high salt diet reversed this difference. While female Sprague Dawley rats had a relatively stable blood pressure on the high salt, male rats displayed a further increase in blood pressure to high salt [90]. Aortic rings from male spontaneously hypertensive rats have a larger vasoconstriction to ANG II than females [91]. In a human cohort, women and men had a similar blood pressure increase in response to ANG II, but men had a higher baseline blood pressure. In addition, ANG II infusion resulted in a reduced glomerular filtration rate in women compared to men, inferring a protection against increased glomerular pressure [86]. The apparent sex difference in RAAS effectors and the response to ANG II warrant a better understanding of various mediators that may be involved in the regulation and response to RAAS in cardio-renal health.

RAAS activation has also been implicated in age-associated arterial pro-inflammation and arterial remodeling [92,93]. Pro-inflammatory overexpression of the effectors in the RAAS pathway, such as ANG II, angiotensinogen, ACE, and angiotensin receptor 1, provoke age-related cardiac and carotid remodeling in rodents [94,95,96]. Upregulation of ANG II, angiotensinogen, ACE, and angiotensin receptor 1 also participate in age-related remodeling of the aortic wall of both humans and nonhuman primates [74,92,97]. Interestingly, several studies from 1977–1994 reported a reduction of renin with age in the plasma of rats and humans [98,99,100,101,102]. This data conflicts with more current reports of elevated ANG II and other RAAS effectors, thus it is important to better understand how all components of RAAS may be impacted by age and how together the age-induced changes may influence cardio-renal health.

RAAS and GPER1 potentially intersect through concurrent effects on ANG II and other downstream pathways. GPER1 has been suggested to impact ANG II effects. In isolated aortic rings from ovariectomized (OVX) mRen2.Lewis rats, a model of estrogen and salt sensitivity, chronic G-1 treatment causes a reduction in ANG II-induced vasoconstriction [103]. In fact, G-1 treatment increased ACE2 expression and decreased ACE expression in these tissues, promoting a reduction in vascular tone [103].

GPER1 has also been implicated in aldosterone-mediated responses. Aldosterone signals its effects via either the mineralocorticoid receptor (MR) or non-MR dependent pathways [104]. Evidence suggests that aldosterone contributes to GPER1 signaling in the vasculature [105,106,107]. However, it is unclear if this effect is through an interaction between the aldosterone signaling and GPER1 or if aldosterone can directly activate GPER1 [108]. GPER1 has been shown to play a role in aldosterone-mediated signaling in the kidney [66,109,110]. Cheng et al. showed that, in human embryonic kidney 293 cells, aldosterone did not bind GPER1 directly [66]. Thus, the effects of aldosterone on renal GPER1 signaling are likely through an interaction of aldosterone and GPER1 signaling pathways [66]. Recent studies report that GPER1 is necessary for non-MR aldosterone dependent pathways [111,112]. Cheng et al. showed that in primary mouse collecting duct cells blockage of GPER1 with G-36 impairs the aldosterone mediated signaling cascade [109]. Gros et al. suggest that aldosterone may be an agonist for GPER1. While they did not test direct binding of aldosterone to GPER1, they used G-1 and G-15 to show that GPER1 is necessary for the non-MR-mediated vascular effects of aldosterone [111]. Additional research demonstrated that GPER1 predominantly stimulates the release of aldosterone in vascular endothelial cells [105,106].

Additional studies are needed to identify the interaction between GPER1 and aldosterone and to determine if this interaction is evident in a specific tissue- or cell-specific manner in males and females. Improving our understanding of how GPER1 impacts the RAAS system, particularly aldosterone non-MR signaling, could provide targets for management of blood pressure and water and electrolyte balance, which are often dysregulated with CV and renal diseases.

#### 2.3.2. Endothelin-1

The endogenous peptide, ET-1 is a potent vasoconstrictor [113]. Within vascular endothelial cells, pre-pro-ET-1 precursor is cleaved in two steps by ET-1 converting enzymes, ECE-1 and ECE-2, to generate active ET-1 [113]. ET-1 actions are mediated by activation of its G protein-coupled receptors, ET_A_ and ET_B_. Within the vasculature, ET_A_ is primarily expressed in VSMCs [114] and mediates vasoconstriction, oxidative stress, and cell proliferation [115,116]. ET_B_ is primarily expressed in vascular epithelial and endothelial tissue as well as VSMCs [114] and mediates ET-1 clearance and vasodilation through the release of NO [115,116,117,118]. Aging [119,120,121,122], menopause [123], and the male sex [124,125] are independently associated with higher plasma ET-1 levels, while the menstrual and pregnancy-induced rise in female gonadal hormones contribute to lowering plasma ET-1 levels [122,124,126,127,128,129].

These correlations invite curiosity for the role of estrogen signaling on the ET-1 system. Given that aging is a risk factor for hypertension [4] and ET-1 plays a critical role in the maintenance of blood pressure, ET-1 changes with age are thought to be in part responsible. Ovarian hormones have been shown to modulate the fluctuations in ET_A_ and ET_B_ responses in young pre-menopausal women [130]. In postmenopausal women, ET_B_-induced vasodilation is impaired compared to young women [131]. Lower serum levels of GPER-1 are associated with hypertension in postmenopausal women [132], but the relationship between GPER-1 and age-related ET_B_ dysfunction is unclear.

ET-1 signaling system has been implicated in GPER1-induced vascular effects. G-1 activation of GPER1 attenuates ET-1-mediated vasoconstriction and causes acute vasodilation in porcine coronary arteries [133]. G-1 increases VSMC [Ca^2+^]_i_ acutely [60] while GPER1 KO reduces VSMC [Ca^2+^]_i_ [134]. Additionally, G-1 reduces Ca^2+^ spikes in VSMC [135]. GPER1 KO mice carotid arteries treated with ET-1 have a greater vasoconstriction response, but a reduced ET-1-stimulated VSMC [Ca^2+^]_i_ increase, suggesting that arteries lacking GPER1 are more sensitive to VSMC [Ca^2+^]_i_ [134]. ET_A_ and ET_B_ expression levels are indifferent in GPER1 KO carotid arteries [134], which pinpoints GPER1 to be a potential regulator of VSMC [Ca^2+^]_i_ in order to balance the vasoconstrictive effects of ET-1.

In aortic tissue of male mice, acute inhibition of GPER1 by G-15 decreased ET_B_ receptor-stimulated NO bioactivity [57]. Overall, these studies suggest that GPER1 displays protective vascular effects. However, some data suggests that GPER1 is required for cardiac aging [136,137]. An aging study comparing 2-year-old mice with age-matched GPER1 KO mice revealed that GPER1 caused the age-dependent upregulation of ET_B_ and ECE-2 gene expression in myocardial tissue [137]. Whether global deletion of GPER1 may have had induced compensatory changes in other sex hormonal signaling pathways remains to be determined.

The renal ET-1 system promotes natriuresis [138,139] and diuresis [140], contributing to sodium homeostasis [141]. Renal ET-1 is most produced in the inner medullary collecting duct cells, which also have the highest expression of ET_A_ and ET_B_, compared to other parts of the kidney [142]. More so, GPER1 receptor expression is greater in female rat kidney tissues, compared to males [56,70]. We recently showed that activation of GPER1 within the renal medulla of female rats evokes natriuresis and diuresis [56,143]. Specifically, we provided genetic and pharmacological evidence that both ET_A_ and ET_B_ collaborate to mediate GPER1-evoked natriuresis and diuresis [56,143]. Furthermore, we showed that female GPER1 KO mice, but not males, have less ET-1, ET_A_, and ET_B_ mRNA expression in kidney tissues [56]. These data suggest that estrogen’s permissive actions on urinary sodium excretion via GPER1 activation is mediated through the renal ET-1 pathway. Sex-specific studies on young and aged hypertensive animal models will help trace the dynamic mechanistic relationship between GPER1 signaling and the ET-1 systems as it relates to sodium homeostasis and blood pressure regulation.

#### 2.3.3. Nitric Oxide

NO is a well-known vasodilator. Three isoforms of NO synthase (NOS) have been identified including: neuronal (nNOS; NOS1), inducible (iNOS; NOS2), and endothelial (eNOS; NOS3) [144]. These three isoforms utilize molecular oxygen, L-arginine, enzymes, and co-factors to generate NO [144,145,146]. NO signals primarily via the activation of soluble guanylyl cyclase and the generation of cyclic guanylyl guanosine monophosphate (cGMP) resulting in vasodilation [144]. Impaired NO signaling has been implicated in the pathophysiology of CV and renal diseases [147,148,149,150]. NO signaling is thought to have sex specific expression differences and may play a role in male–female differences in cardio-renal health. NO release in response to acetylcholine is elevated in the aorta of female rats compared to males [151,152]. Similarly, eNOS expression is higher in the kidney of female rats compared to males [153,154].

Several studies have suggested that NO production and release are reduced with age in rodents [155,156]. Urinary excretion of NO metabolites, nitrite and nitrate, are reduced in aged rats compared to young [155,156]. Similarly, the NO substrate L-arginine is reduced in the serum of aged rats compared to young [156]. The apparent decrease in NO level, inferred by reduced urinary excretion of NO metabolites and serum L-arginine, coincides with renal injury progression. Aged rats exhibit a reduced glomerular filtration rate and an increased renal vascular resistance and glomerular sclerosis [155,156]. In addition, eNOS and nNOS expression decrease in the aging male rat kidney [156,157,158].

NO bioavailability and activity are reduced in human populations with aging [159,160,161]. In a human study, vasodilation to acetylcholine in the brachial artery was reduced with age [161]. Asymmetric dimethylarginine (ADMA), an endogenous NOS inhibitor, is increased in plasma in aged humans compared to young [159]. The increase in ADMA as well as the reduction in L-arginine are suspected to reduce NO bioavailability with age thus reducing NO-mediated vasodilation, potentially participating in the increased blood pressure associated with aging [159,160,161].

Evidence suggests that GPER1 stimulates NO production and promotes relaxation in multiple vascular beds [162,163,164,165,166]. GPER1 activation by G-1 increases NO in endothelial cells leading to vascular relaxation [162,164,165]. Similarly, GPER1 expression is reduced in mesenteric arteries in aged female and young male Lewis rats compared to young female rats. This downregulation is associated with reduced vasorelaxation to estrogen and G-1 [167]. Moreover, GPER1 deletion in female mice attenuates NO bioactivity in carotid arteries [168] and promotes vasoconstriction of aortic rings in male mice [57]. Additionally, GPER1 KO promotes atherosclerosis in the aorta of female mice [168]. Several studies suggest that GPER1 is responsible for acute estrogen-dependent vasodilation in both male and female rat aortas, which involved an increase in NO levels and a decrease in blood pressure [169,170,171] possibly through mediation of eNOS-dependent NO formation [164,167]. Contrary to the studies above, Meyer et al. showed that GPER1 deletion in male mice attenuates age-related impairment of NO-mediated relaxation [172]. The majority of the literature support GPER1 mediation of NO bioavailability as a potential protective mechanism in the vasculature, particularly in the cardio-renal systems. The interactions of NO and GPER1 could provide additional novel insights into physiologic and pathological changes in cardio-renal health with age and sex.

#### 2.3.4. Reactive Oxygen Species

ROS constitute essential components of aerobic respiration and cell signaling and are delicately balanced with antioxidant systems under physiological conditions. Nevertheless, excess production of ROS, including superoxide (O_2_^−^), hydroxyl radicals and hydrogen peroxide, leads to cellular toxicity in various disease states such as atherosclerosis, coronary ischemia, cardiac hypertrophy, and heart failure [173] as well as acute kidney injury (AKI) [174] and CKD [175]. Interestingly, there are sex differences in ROS profiles. Several studies in rats, humans, and human derived VSMCs showed that oxidative stress and markers such as lipid peroxidation are elevated in males compared to females [176,177,178,179]. In addition to baseline differences there are apparent differences in oxidative stress in pathological settings. Female spontaneously hypertensive rats (SHR) exhibit a reduced blood pressure and reduced markers of oxidative stress compared to their male littermates [180]. Similarly, ANG II induces hypertension and oxidative stress to a lesser degree in female compared to male SHR [181]. Combined these studies indicate the importance of understanding how sex may influence oxidative stress. Moreover, progressive accumulation of oxidative damage is hypothesized to be the primary mechanism of aging, termed as oxidative stress theory of aging [182]. Thus, potential antioxidant mechanisms to target these processes are major topics under investigation [183].

Emerging evidence suggests that GPER1 signaling regulates oxidative stress, a major player in the pathogenesis of aging and cardiovascular and renal disease [175,184]. Several studies point to GPER1 as a crucial regulator of oxidative response. GPER1 is hypothesized to mediate the cardioprotective effects of estrogen through modulation of oxidative stress. In the mRen2 female rat model, GPER1 activation with G-1 is protective against salt-induced aortic remodeling. This protective effect is independent of changes in blood pressure, but associated with reduced lipid peroxidation, a marker of oxidative stress [185]. In the same rat model, treatment with G-1 also significantly attenuates proteinuria and lipid oxidation in the renal cortex [67]. Bopassa et al. found that G-1 reduces myocardial infarct size and improves heart function after ischemia-reperfusion injury by inhibiting mitochondria permeability transition pore opening [186]. Acute oxidative stress opens the mitochondrial permeability transition pore, leading to cellular death, while inhibition of mitochondrial permeability transition pore opening confers protection against ischemic cardiac injury [186].

In both male and female mice, KO of GPER1 results in left ventricular dysfunction and remodeling [65], whereas pharmacological GPER1 activation mitigates the adverse effects of estrogen loss on left ventricular function in OVX rats [187]. Female GPER1 KO mice were also found to exhibit increased oxidative damage in cardiomyocytes and upregulation of oxidative stress-related genes along with cardiac dysfunction [188]. In turn, treatment with a mitochondria-targeted antioxidant effectively ameliorates cardiac dysfunction, suggesting that the cardioprotective effects of GPER1 may be mediated by its antioxidant action [188]. GPER1 activation has also been reported to provide renoprotection against methotrexate-induced oxidative stress in human renal epithelium cells [189], neuroprotection against hydrogen peroxidase-induced apoptosis in primary mouse cortical neuronal culture [190] and cardio protection against doxorubicin-induced oxidative injury in male rats [191]. Furthermore, in a female mouse model of diabetes, activation of GPER1 attenuated oxidative stress in pancreatic β cells and promotes islet cell survival [73].

On the contrary, there are a few studies suggesting that GPER1 may induce oxidative stress and cardiovascular toxicity. A study by Meyer et al. investigating genetic (GPER1 KO) or pharmacologic (G-36) inactivation of GPER1 in male mice suggests that GPER1 may play a role in cardiovascular aging and disease through promotion of ROS [136]. Vascular aging is characterized by increased O_2_^−^ formation that impairs NO-dependent vasorelaxation [136]. Meyer et al. showed that aged GPER1 KO mice have less O_2_^−^ formation in VSMC and were protected from age-related vasodilatory impairment compared to wild type aged mice. This is likely through a reduced NADPH oxidase 1 in response to GPER1 deletion [136]. Further, Meyer et al. demonstrated that GPER1 KO prevented endothelial aging in renal arteries of male mice [172]. Although abundant evidence suggests an important relationship between GPER1 and oxidative stress, the specific role of GPER1 activation in different pathophysiologic conditions and whether this role changes with increasing age remain to be investigated. Further research is necessary to clarify the role of GPER1 in regulating ROS and mitochondrial function. Understanding the mechanisms by which GPER1 could impact ROS could provide additional targets for therapeutics in the treatment of CV and renal diseases.

#### 2.3.5. Immunity and Inflammation

In the last decades, chronic inflammation has been acknowledged as a major driver of cardiovascular and renal disease [192,193,194]. Indeed, inflammation is the earliest step in the pathogenesis of atherosclerosis [195] and increased inflammatory cytokines are consistent predictors of the development and progression of cardiovascular disease [196]. Similarly, patients with CKD suffer from a chronic inflammatory state, which is associated with worse prognosis of the disease [197]. Furthermore, inflammation plays a major role in aging, a condition often referred to as inflammageing [198]. Indeed, inflammageing, characterized by a chronic pro-inflammatory state, affects almost all physiological systems and strongly correlates with cardiovascular and renal comorbidities [199].

GPER1 signaling appears to have an important role in regulation of inflammatory response in various systems, including the vasculature [200,201], adipose tissue [202], gastrointestinal tract [203,204], and central nervous system [205], as well as in pathologies such as cardiovascular disease [206], nephrotoxicity [189], and cancer [206]. Moreover, GPER1 mRNA is expressed in both early and mature cells of the immune system, suggesting that GPER1 has a functional role in maturation and function of immune cells [207]. GPER1 works to induce the expression of anti-inflammatory cytokine interleukin 10 and expand the regulatory T cell population [208,209], whereas GPER1 deficiency results in a pro-inflammatory state in animal models [202]. The different roles of GPER1 activation in each immune cell type have been discussed in depth elsewhere [207]. Overall, GPER1-induced immune modulation may have important implications for CV and renal health that should be noted.

GPER1 appears to mediate anti-inflammatory effects in the vasculature and cardiac tissue. G-1 was shown to ameliorate tumor necrosis factor-induced upregulation of pro-inflammatory mediators in the endothelium in vitro [200]. Pre-treatment with G-1 also attenuates C-reactive protein-induced inflammatory response in macrophages and VSMC derived from young but not old female mice, suggesting that GPER1′s actions may be age-dependent [210]. Meyer et al. found that deletion of GPER1 in female mice leads to increased atherosclerosis progression and plasma cholesterol levels along with vascular inflammation, observed as a striking increase in macrophage and T cell accumulation in the aortic root [168]. Curiously, immune cell accumulation increases to the same extent with deletion of GPER1 or OVX, signifying that GPER1 was entirely responsible for the anti-inflammatory effect of estrogen. Moreover, treatment of OVX mice with G-1 attenuates immune cell infiltration and atherosclerosis [168], suggesting that GPER1 activation may be a therapeutic approach to target the increased cardiovascular risk in postmenopausal women. Wang et al. elucidated a role of GPER1 signaling on NLR Family Pyrin Domain Containing protein 3 (NLRP3) inflammasome activity. They found that cardiomyocyte-specific GPER1 deletion in mice is associated with upregulation of NLRP3 inflammasome-related genes. Importantly, NLRP3 inhibition significantly improves heart function ameliorated by GPER1 deletion [211], suggesting that GPER1 signaling may be crucial to regulate NLRP3 inflammasome in cardiac tissue.

GPER1 activation appears to have anti-inflammatory effects in the renal system as well [212]. Estrogen has been proposed to be protective against kidney disease in premenopausal women [56]. GPER1 appears to also play a role in estrogen-mediated cytoprotection due to its high expression in the renal tubular cells [56]. Using methotrexate-induced nephrotoxicity models in human renal epithelial cells, Kurt et al. showed that co-incubation of cells with either E_2_ or G-1 reduces the high levels of interleukin-1β and interleukin-6 induced by methotrexate [189]. However, our studies indicate that although G-1 treatment effectively prevented salt-induced proteinuria and proximal tubular injury in female Dahl salt sensitive rats, it did not decrease salt-induced immune cell accumulation in the kidney [213]. These results highlight the need for further research to confirm the immune effects of GPER1 activation in cardiovascular and renal systems and to elucidate the potential role of GPER1 signaling in inflammageing. A better understanding of the mechanisms by which GPER1 could impact inflammation and immune infiltration could provide additional targets for therapeutics in the treatment of CV and renal diseases.

Overall, GPER1 signaling has a clear role in RAAS, ET-1, NO, ROS, mitochondria, immunity, and inflammation signaling pathways that play critical roles in cardio-renal health (Figure 3). These interactions are not completely understood, but each present an opportunity for improvement of cardio-renal health.

## 3. GPER1 Signaling in Cardiovascular and Renal Disease

Several lines of research implicate an important role for GPER1 signaling pathways in different cardiorenal disease pathologies. In the current section, we will highlight the potential role of GPER1 in hypertension, cardiac and kidney diseases.

### 3.1. Hypertension

Hypertension, also known as the “silent killer”, is a leading risk factor for cardiovascular disease. It is responsible for about 9.5 million deaths per year worldwide [214,215]. According to Centers for Disease Control and Prevention (CDC) data, the prevalence of hypertension is higher in the older population (>60 years old, 74.5%). While the overall prevalence of hypertension in younger adults (18–39 years old) is 22.4%, there is a higher prevalence of hypertension in men (31.2%) than in women (13.0%) [216]. This sex difference decreases with age, with a similar prevalence in men and women by the age of 60 [216]. Risk factors of hypertension include unhealthy diets (high sodium or low potassium), physical inactivity, consumption of tobacco and alcohol, and obesity [217]. Despite the proven pervasiveness of hypertension particularly among older adults, many challenges remain today in the treatment and prevention of hypertension clinically.

Studies point to GPER1 as a novel regulator for blood pressure and sodium homeostasis. GPER1 activation with G-1 acutely reduces blood pressure in male Sprague Dawley rats [60]. Chronic systemic treatment with G-1 also lowers blood pressure in OVX Sprague Dawley rats [56] and OVX mRen2.Lewis rats [103]. In addition, increased blood pressure is observed in female GPER1 KO mice by 9 months of age [48] and women with a hypoactive GPER1 variant [218]. GPER1′s blood pressure lowing actions are multifactorial and involve regulating the vascular tone. G-1 promotes vasodilation in different vascular beds, including coronary arteries [60,133], carotid arteries [133], aorta, and mesenteric arteries [170]. Dilation downstream of GPER1 activation involves multiple signaling pathways including NO [162,166,167,169], ET-1 system [133], and the RAAS pathway [103,105,106,107,111,112], depending on the capillary bed and species. Determining whether G-1-induced vasodilation is sex-specific has been controversial. Debortoli et al. showed greater G-1-induced vasodilation response in female Wister rats compared to males [219] and in isolated arteries from postmenopausal women compared to age-matched men [220]. However, other studies have shown no sex differences in GPER1-induced vasodilation in rat mesenteric arteries [165,221], carotid arteries [171], and cerebral arterioles [222].

Recently, we showed that renal medullary GPER1 activation promotes ET-1 dependent natriuresis in female, but not male, rats, which may also contribute to GPER1 blood pressure lowering actions [56,143]. GPER1-mediated estrogen action along the nephron is still unclear. In distal convoluted tubule cells, estrogen increases NaCl cotransporter phosphorylation [223]. Of note, GPER1 is required for E_2_-induced spike in Ca^2+^ in the distal renal tubules [224]. Importantly, GPER1 is heavily expressed along the distal convoluted tubule cells basolateral membrane [223].

In women, GPER1 hypofunctional variant is associated with increased blood pressure [218] and increased plasma low-density lipoprotein cholesterol [225]. In addition, a study by Liu et al. suggested that serum GPER1 is a protective factor against hypertension in menopausal women, but not premenopausal women [132]. Improving our understanding about signaling pathways mediating GPER1′s blood pressure lowering actions is clinically important for drug development, especially for preventing postmenopausal hypertension.

### 3.2. Heart Disease

Heart disease is the number one leading cause of death in the US [226]. CDC data reported 696,962 deaths from heart disease in 2020, which is about 1 in 4 deaths [2]. According to the data of the American Heart Association, the number of men with a diagnosed heart attack or fatal coronary heart disease (CHD) is higher than women [4]. In addition, the prevalence of heart attack and CHD increases with aging, independent of sex [4]. Risk factors for heart disease include high blood pressure, dyslipidemia, diabetes mellitus, and obesity [4,227]. Advanced measures are required to avoid heart disease and provide better care for the impacted population.

Previous studies have provided evidence for the cardioprotective role of GPER1 activation against heart failure in male mice [228] and myocardial inflammation in male SHR [76]. In addition, it was reported that GPER1 activation mitigates doxorubicin-induced cardiotoxicity in male rats, [191], protects against cardiomyocyte death [64,229], improves myocardial mechanical performance, and reduces infarct size in isolated rat and mouse hearts after ischemia reperfusion injury through the involvement of PI3K kinase/AKT signaling pathway [72,230,231,232]. Further, GPER1 deletion in female mice led to cardiac remodeling and oxidative stress [188]. G1 decreases cardiac remodeling induced by salt [233] and hypertrophic regulators like ANG II and ET-1 [77]. G-1 also inhibits cell cycle gene expression including cyclin B1 and CDK1 which are involved in cardiac fibroblast and mast cell proliferation and interstitial remodeling [77]. However, the exact signaling actions and transduction pathways of cardiac GPER1 are not completely understood.

Estrogen-induced regulation of heart rate appears to be linked to GPER1 rather than ERα and Erβ [234,235]. In addition, GPER1 has been shown to modulate maternal estrogen levels in zebrafish, which are essential for appropriate embryonic heart rates [235]. Overall, GPER1 signaling is important in maintenance of cardiac health and a potential therapeutic target in heart disease.

### 3.3. Kidney Disease

Acute kidney injury (AKI) is a pathological condition characterized primarily by a rapid increase in serum creatinine over a short period of time [236]. AKI causes around 1.7 million global deaths annually and affects more than 13.3 million people globally [237]. Men have a higher incidence of AKI-dialysis compared to women [238]. Further, aging is associated with increased incidence of AKI [239,240,241]. Risk factors for AKI include kidney ischemia reperfusion and reduced blood flow leading to renal hypoxia [242]. Renal hypoxia can lead to tissue damage [242,243] and this can cause fluid and waste built up resulting in life-threatening complications, or even death if left untreated [244]. Therefore, more preventive measures are required to avoid complications and provide better care for people impacted by AKI.

Previous studies have reported an important role of E_2_ signaling in the regulation of eNOS expression in rat kidneys, which provides protection against ischemia reperfusion-induced AKI [245]. Moreover, Wu et al. reported that E_2_ accelerates the regeneration of renal tubules in male rats after ischemia reperfusion-induced AKI through reducing inflammation [246]. Furthermore, GPER1 has been shown to have protective effects against ischemia reperfusion injury in the rat and human heart [64,186,232,247]. To date, the contribution of GPER1 in AKI is not completely clear, however, a better understanding of its role could reveal novel avenues for therapeutic intervention.

Damage in the structure and/or function of the kidney often leads to CKD, which is characterized by its irreversibility and slow progression [5]. An estimated 15% of US adults (37 million people) have CKD [9]. CKD is more common in the older population (>65 years old, 38.1%) than the younger population (18–44 years old, 6.0%). CDC data reports a slightly higher percentage of CKD prevalence in women (14.3%) than men (12.4%) during their life span [9].

Interestingly, GPER1 activation has been shown to reduce kidney damage in mRen2.Lewis rats that were fed a high-salt diet [67]. Earlier studies reported that ovariectomy increases levels of renal inflammatory markers in Dahl salt-sensitive female rats [248], hinting at the protective role of the female sex hormone against hypertension and end-organ kidney damage. Moreover, we reported that G-1 ameliorates kidney damage in high salt-fed female Dahl salt-sensitive via preservation of the proximal tubule brush border integrity [213]. The definitive role of GPER1 in CKD initiation and progression is not entirely known yet, but evidence supports its role as a protective mechanism against kidney damage, independent from blood pressure changes.

Collectively, evidence suggests GPER1 is protective in hypertension, heart, and kidney disease (Figure 3). A more in depth understanding of the complex pathway interactions and signaling mechanisms are therefore imperative in furthering cardio-renal health. In addition, cardio-renal diseases often share many risk factors and comorbidities therefore understanding the complex interactions and overlap of these signaling pathways and GPER1′s potential role could lead to benefits in many aspects of human health. One such avenue is in cancer research. Patients with CV and renal diseases have a higher incidence of malignancies [249,250,251,252,253,254,255,256,257]. Conversely, studies point to increased risk for CV and renal complications in cancer survivors [258,259,260,261]. In addition, some therapeutic regimens used for treatment of breast and prostate cancer pose potential cardiac and nephrotoxic risk [260,262]. Given the role of GPER1 signaling in cardiorenal disease and carcinogenesis, improving our understanding of the cardiovascular and renal responses to GPER1 will open another avenue to improve the care provided to cancer patients.

## 4. Future Directions and Translational Perspective

Future studies aimed at better understanding the impact of GPER1 on cardio-renal disease mechanisms during aging could reveal novel therapeutics aimed at improving clinical outcomes. RAAS, NO, ET-1, ROS, inflammation, and immune infiltration have all been implicated in CV and renal disease. Furthermore, there have been described changes in each of these pathways with age and sex bolstering support for understanding mechanisms that could be impacted by sex hormones. GPER1 is one such mechanism that could provide protection against age-related CV and renal disease development and progression. Recently Sharma et al. investigated the preclinical potential of G-1 in mouse models of obesity and diabetes [263]. While there may be a therapeutic promise for G-1 in metabolic disease, it is unknown if this promise could extend to cardio-renal disease. In clarifying the mechanisms by which GPER1 interacts with these signaling systems, novel therapeutics for management of cardiovascular and renal disease could be developed. Ultimately additional work needs to be done still to fully understand the potential of GPER1 in regulating mechanisms of cardio-renal health in elderly.

## Figures and Tables

**Figure 1 biomolecules-12-00412-f001:**
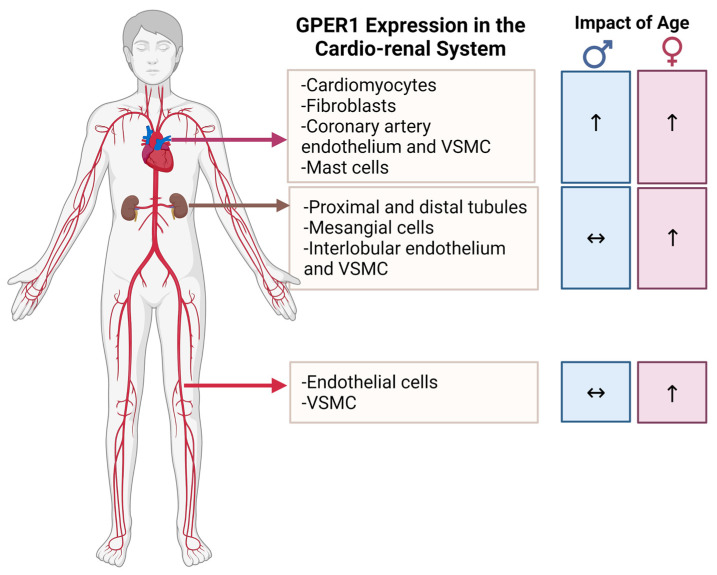
GPER1 expression in the cardio-renal system; impact of age. Abbreviations: GPER1—G protein-coupled estrogen receptor 1. VSMC—Vascular smooth muscle cell. Created with BioRender.com, accessed on 28 January 2022.

**Figure 2 biomolecules-12-00412-f002:**
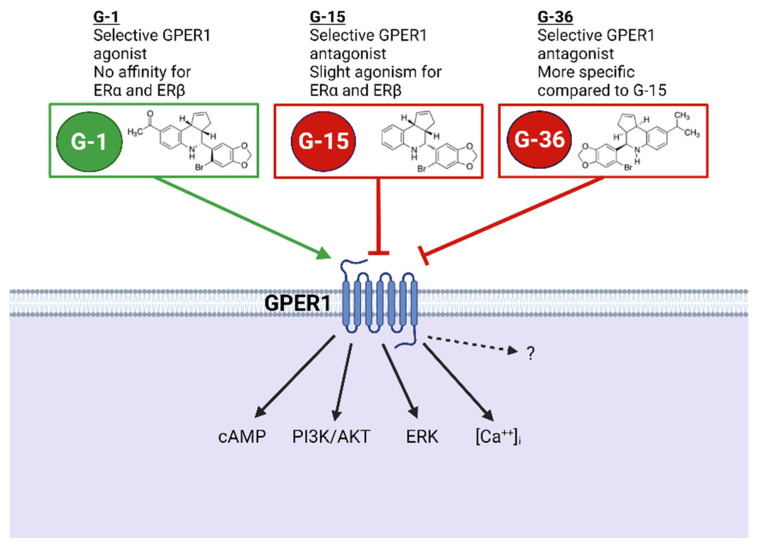
Pharmacological tools to study GPER1 function. Abbreviations: GPER1—G protein-coupled estrogen receptor 1. ERα/ERβ—Estrogen receptor alpha/beta. CAMP—Cyclic adenosine monophosphate. PI3K—Phosphoinositide 3-kinase. AKT—Protein kinase B. ERK—Extracellular signal-regulated kinases. [Ca^2+^]_i_—Intracellular calcium. Created with BioRender.com, accessed on 23 February 2022.

**Figure 3 biomolecules-12-00412-f003:**
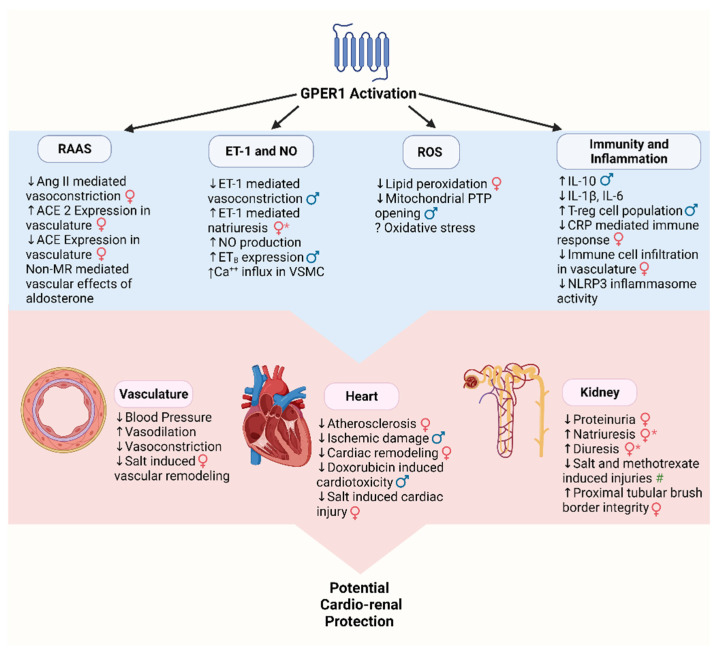
Mediators of GPER1 actions within the cardiovascular and renal systems. Observations reported in only one sex are denoted with the corresponding sex symbol. Observations reported in both sexes have no special designation. * Denotes sex-specific observations. # Denotes observations with unspecified sex. Abbreviations: GPER1—G protein-coupled estrogen receptor 1. Ang II—Angiotensin II. ACE—Angiotensin converting enzyme. MR—Mineralocorticoid receptor. ET-1—Endothelin 1. NO—Nitric oxide. ET_B_—Endothelin receptor B. Ca^2+^—Calcium. VSMC—Vascular smooth muscle cell. PTP—Permeability transition pore. IL—Interleukin. T-reg—Regulatory T cell. CRP—C reactive protein. NLR3P—NLR family pyrin domain containing 3. Created with BioRender.com, accessed on 23 February 2022.

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
