# Peer review of "Emerging Roles for G Protein-Coupled Estrogen Receptor 1 in Cardio-Renal Health: Implications for Aging"

_biomolecules, 2022, doi:10.3390/biom12030412_

Round 1

Reviewer 1 Report

The review entitled “Emerging Roles for G Protein-coupled Estrogen Receptor 1 in Cardio-renal Health: Implications for Aging” by Singh et. al. describes the current literature regarding GPER1 and cardio-renal health. The authors also described and summarized the existing literature in the aging populations (human and animal studies). In general the review is very carefully written and considerable amount of literature search has been done to prepare it.

My comments are listed below:

Major

  1. In lines 187-188, the authors are describing the sex differences in the response to Ang II stating that “More importantly, there is a sex difference in the response to ANG II. Several studies 187 show female mice have a reduced response to ANG II induced blood pressure increase 188 [88,89]”. The concept is controversial. In 2008, a study by Sartori and colleagues pointed out that in the absence of a functional RAAS, the blood pressure response to Ang II is higher in females than in males. Please include an explanation for this controversy in this paragraph.
  2. Since the review is highlighting the implications for aging in the title, It would be beneficial if the authors could include a table to discuss and summarize the role for GPER1 in aging via controlling/interacting with the listed blood pressure and cardio-renal health modulating mechanisms (RAS, ET, inflammation, etc.).

Minor:

  1. In line 245; “Adult aging” should be changed to “Aging” only
  2. In line 176; “RAAS system has” should be changed to “RAAS has”, s in RAAS already denotes system.
  3. Line 478; word in is doubled.
  4. Line 479; “treatment of G-1” should be changed to “treatment with G-1”.
  5. Line 496; “In distal convoluted tubule cells estrogen, increases” , the comma should be placed before the word estrogen,
  6. In figure 3, the abbreviation VSCM should be VSMC.

Author Response

Reviewer 1:

The review entitled “Emerging Roles for G Protein-coupled Estrogen Receptor 1 in Cardio-renal Health: Implications for Aging” by Singh et. al. describes the current literature regarding GPER1 and cardio-renal health. The authors also described and summarized the existing literature in the aging populations (human and animal studies). In general the review is very carefully written and considerable amount of literature search has been done to prepare it.

Major Comments:

  1. In lines 187-188, the authors are describing the sex differences in the response to Ang II stating that “More importantly, there is a sex difference in the response to ANG II. Several studies 187 show female mice have a reduced response to ANG II induced blood pressure increase 188 [88,89]”. The concept is controversial. In 2008, a study by Sartori and colleagues pointed out that in the absence of a functional RAAS, the blood pressure response to Ang II is higher in females than in males. Please include an explanation for this controversy in this paragraph.

Response:

Thank you for bringing this manuscript to our attention. We have cited the study and addressed the findings in line 189-193. These findings are interesting as they implicate complex interactions among various parts of the RAAS in the sex specific presentations of ANG II induced hypertension. The study further supports our conclusion in lines 198-201 that a better understanding of the RAAS and various mediators that may be involved in sex differences is imperative.

  1. Since the review is highlighting the implications for aging in the title, It would be beneficial if the authors could include a table to discuss and summarize the role for GPER1 in aging via controlling/interacting with the listed blood pressure and cardio-renal health modulating mechanisms (RAS, ET, inflammation, etc.).

Response:

Thank you for this suggestion. We did go through the review and attempt to create a table summarizing the role of GPER1 in aging. Unfortunately, there is very little literature on the effects of GPER1 in aged models. The purpose of this review is to explore the current literature on GPER1 and the many signaling pathways that could be interacting with GPER1 in cardio-renal health during aging. There is a particular interest in aging as these pathways are altered with age and can be affected by the sex hormone changes that are also seen in age. Further, GPER1 effects on the discussed signaling pathways and organ systems are illustrated in figure 3.

Minor:

  1. In line 245; “Adult aging” should be changed to “Aging” only
  2. In line 176; “RAAS system has” should be changed to “RAAS has”, s in RAAS already denotes system.
  3. Line 478; word in is doubled.
  4. Line 479; “treatment of G-1” should be changed to “treatment with G-1”.
  5. Line 496; “In distal convoluted tubule cells estrogen, increases” , the comma should be placed before the word estrogen,
  6. In figure 3, the abbreviation VSCM should be VSMC.

Response:

Thank you for identifying these textual errors. We have altered each mistake accordingly.

Reviewer 2 Report

The review by Singh et al. comprehensively summarizes the function of G protein-coupled estrogen receptor 1 and the current understaning on the mechanim of action in impacting cardio-renal diseases. This review is well-organized and written. Several points could be added as listed below:

  1. In introduction section, the papers referred were from 2019. Could the authors replace the statistics with the most recent reports?
  2. Sex hormone is highly related to hormone-meditaed cancer incidence. Could the authors briefly summarize the incidential relationship between cardio-renal disease and cancers (particulay breast and prostate cancer)?
  3. Figure 2 needs to be improved. First, could the authors add the chemical structure of the 3 compounds? Second, the downstream components of GPER1 need to be illustrated. Are there any studies using G15, G1, G36 in cardio-renal disease models?
  4. In Figure 3, it would be great if the authors could highlight which properties are gender specific. Since many of these changes are not applied in both male and female.
  5. How about the clinical application potential of GPER1 agonists or antagonist? It would be good if the authors can give a summary and perspective. 

Author Response

Reviewer 2:

The review by Singh et al. comprehensively summarizes the function of G protein-coupled estrogen receptor 1 and the current understanding on the mechanism of action in impacting cardio-renal diseases. This review is well-organized and written. Several points could be added as listed below:

  1. In introduction section, the papers referred were from 2019. Could the authors replace the statistics with the most recent reports?

Response:

Thanks for pointing this out. We did update the statistics based on the most recent data available from the National Center for Health Statistics (2020 death data).

  1. Sex hormone is highly related to hormone-mediated cancer incidence. Could the authors briefly summarize the incidental relationship between cardio-renal disease and cancers (particularly breast and prostate cancer)?

Response:

We thank the reviewer for the suggestion. We have added a section regarding cardio-renal disease and cancer that can be found in lines 578-587

  1. Figure 2 needs to be improved. First, could the authors add the chemical structure of the 3 compounds? Second, the downstream components of GPER1 need to be illustrated. Are there any studies using G15, G1, G36 in cardio-renal disease models?

Response:

Thank you for your suggestions. We have added the structures of G-1, G-15, and G-36 to the figure as recommended. We have the downstream components listed in figure 2. Further, figure 3 provides more details about the downstream effectors impacted by GPER1 in each of the discussed signaling pathways and organ systems, so we did not represent that again in figure 2. Further, there are multiple studies using GPER1 agonist and antagonists in different cardiorenal disease models (including SHR, Dahl SS rats, IR models, etc). We discussed these studies throughout the manuscript (lines 216-218, 229-230, 231-233, 262-266, 272-273, 284-289, 322-326, 360-364, 364-366, 386-387, 423-424, 432-433, 444-447, 447-450, 486-488, 491-493, 496-498, 526-528, 531-532,532-534, 570-572).

  1. In Figure 3, it would be great if the authors could highlight which properties are gender specific. Since many of these changes are not applied in both male and female.

Response:

Thank you for the suggestion. We have added sex indications to figure 3 as suggested. While some responses are sex specific, many studies were only conducted in one sex or in both sexes, so we have added identifiers and an explanation in the figure legend to clarify that to the reader.

  1. How about the clinical application potential of GPER1 agonists or antagonist? It would be good if the authors can give a summary and perspective. 

Response:

Thank you for this suggestion. We have expanded the “Future Perspective” section to address this point (lines 596-598).

Reviewer 3 Report

Singh R et al. propose an interesting manuscript about the potential role of the GPER-1 receptor in cardiovascular and renal function and its antiaging action. This review titled "Emerging Roles for G Protein-coupled Estrogen Receptor 1 in 2 Cardio-renal Health: Implications for Aging" covers in a well-structured way a comprehensive description of the structure, function, and localization of the GPER-1 receptor, as well as its involvement in different molecular pathways and cardiovascular pathologies and renal disease. Furthermore, the authors have used an appropriate bibliography regarding the number of citations and their relevance, providing a good justification for the information provided in the different sections described in the text.  

Author Response

Reviewer 3:

Singh R et al. propose an interesting manuscript about the potential role of the GPER-1 receptor in cardiovascular and renal function and its antiaging action. This review titled "Emerging Roles for G Protein-coupled Estrogen Receptor 1 in 2 Cardio-renal Health: Implications for Aging" covers in a well-structured way a comprehensive description of the structure, function, and localization of the GPER-1 receptor, as well as its involvement in different molecular pathways and cardiovascular pathologies and renal disease. Furthermore, the authors have used an appropriate bibliography regarding the number of citations and their relevance, providing a good justification for the information provided in the different sections described in the text.  

Response:

We thank you for your thoughtful review and kind words!

Reviewer 4 Report

The review paper by Singh et al., is quite interesting and comprehensive,

Minor points that need to be addressed:

  • L 173 "affects" instead of "effects"
  • L296-L299 wrong font
  • L315-L318 wrong font
  • L326-L331 wrong font
  • L332 remove "mitochondria" since the previous title is : Mediators of GPER1 actions..."
  • L333-L393 superoxide anion (O2-). Please correct the wrong indications
  • L394 The previous general title is "Mediators of GPER1 actions..." but immunity and inflammation are not mediators. The title needs to be changed.
  • L413 remove "in"

Author Response

Reviewer 4:

The review paper by Singh et al., is quite interesting and comprehensive,

Minor points that need to be addressed:

  • L 173 "affects" instead of "effects"
  • L296-L299 wrong font
  • L315-L318 wrong font
  • L326-L331 wrong font
  • L332 remove "mitochondria" since the previous title is : Mediators of GPER1 actions..."
  • L333-L393 superoxide anion (O2-). Please correct the wrong indications
  • L394 The previous general title is "Mediators of GPER1 actions..." but immunity and inflammation are not mediators. The title needs to be changed.
  • L413 remove "in"

Response:

Thank you for pointing out these textual errors. We have revised the text to correct these issues. Regarding the title ROS and Mitochondria, we have removed mitochondria as suggested. Regarding “immunity and inflammation” as a title, we do see your point that immunity and inflammation are not technically mediators. However, the title is intended as a broad category that contains mediators such as interleukins and immune cells.